# Engineering and Characterization of Oncolytic Vaccinia Virus Expressing Truncated Herpes Simplex Virus Thymidine Kinase

**DOI:** 10.3390/cancers12010228

**Published:** 2020-01-17

**Authors:** S. M. Bakhtiar Ul Islam, Bora Lee, Fen Jiang, Eung-Kyun Kim, Soon Cheol Ahn, Tae-Ho Hwang

**Affiliations:** 1Medical Research Center, School of Medicine, Pusan National University, Yangsan 50612, Korea; smbakhtiar@pusan.ac.kr (S.M.B.U.I.); ekkim@bionoxx.com (E.-K.K.); 2Department of Microbiology and Immunology, School of Medicine, Pusan National University, Yangsan 50612, Korea; ahnsc@pusan.ac.kr; 3Department of Pharmacology, School of Medicine, Pusan National University, Yangsan 50612, Korea; bora.lee@bionoxx.com; 4Bionoxx Inc., Seongnam-si, Gyeonggi-do 13554, Korea; jiangfen@bionoxx.com; 5School of Pharmaceutical Science (Shenzhen), Sun Yat-sen University, Guangzhou 510275, China

**Keywords:** oncolytic viruses, vaccinia virus, herpes simplex virus type 1, thymidine kinase, ganciclovir

## Abstract

Oncolytic viruses are a promising class of anti-tumor agents; however, concerns regarding uncontrolled viral replication have led to the development of a replication-controllable oncolytic vaccinia virus (OVV). The engineering involves replacing the native *thymidine kinase* (*VV-tk*) gene, in a Wyeth strain vaccinia backbone, with the herpes simplex virus *thymidine kinase* (*HSV-tk*) gene, which allows for viral replication control via ganciclovir (GCV, an antiviral/cytotoxic pro-drug). Adding the wild-type *HSV-tk* gene might disrupt the tumor selectivity of *VV-tk* deleted OVVs; therefore, only engineered viruses that lacked tk activity were selected as candidates. Ultimately, OTS-412, which is an OVV containing a mutant *HSV-tk*, was chosen for characterization regarding tumor selectivity, sensitivity to GCV, and the influence of GCV on OTS-412 anti-tumor effects. OTS-412 demonstrated comparable replication and cytotoxicity to VV^tk-^ (control, a *VV-tk* deleted OVV) in multiple cancer cell lines. In HCT 116 mouse models, OTS-412 replication in tumors was reduced by >50% by GCV (*p* = 0.004); additionally, combination use of GCV did not compromise the anti-tumor effects of OTS-412. This is the first report of OTS-412, a *VV-tk* deleted OVV containing a mutant *HSV-tk* transgene, which demonstrates tumor selectivity and sensitivity to GCV. The HSV-tk/GCV combination provides a safety mechanism for future clinical applications of OTS-412.

## 1. Introduction

Although scientists discovered that viruses can kill cancer cells over a century ago, interest in oncolytic virus research has been slow to develop until recently [1]. In the past 20 years, substantial breakthroughs in oncolytic virus engineering have been made; in 2015, the US Food and Drug Administration (FDA) and European Medicines Agency (EMA) approved their first oncolytic virus, T-VEC, a genetically engineered herpes simplex virus, for the treatment of advanced melanoma [2]. Meanwhile, many new candidates engineered from other virus families, such as VACV and adenovirus, have entered clinical trials [3,4,5]. In particular, the growing trend of combining oncolytic viruses with other cancer treatments, such as chemotherapies [6] and immune checkpoint inhibitors [7], has sparked significant interest in oncolytic virus research.

Vaccinia virus (VACV) is one of the more preferred backbones for oncolytic virus engineering based on its long history of use as the routine vaccine for smallpox [8]. Current clinical data suggest that the efficacy of oncolytic vaccinia virus (OVV) treatment tends to be dose-dependent [9]. Therefore, in clinical settings, a high treatment dosage (thousands of times higher than vaccination dose) is more likely to be chosen to maximize the OVV’s anti-tumor effects. However, high-level, persistent viral replication can be cause for major safety concerns when dealing with replication-competent oncolytic viruses, including OVV, as addressed by the US FDA in the guidance of Preclinical Assessment of Investigational Cellular and Gene Therapy Products (November 2013).

Regarding these concerns, a safety mechanism for viral replication control might be necessary for human studies. The current US FDA-approved first-line treatments for undesired VACV infection include Vaccinia Immune Globulin Intravenous (VIGIV, approved in 2005) and tecovirimat (approved in 2018), but these drugs are not readily available in many countries outside of the US. The other two second-line treatments, cidofovir and its prodrug brincidofovir, were initially developed to target other viral infections (e.g., cytomegalovirus, adenovirus, and Ebola virus) [10]. All in all, clinical experience with the abovementioned drugs for specifically treating VACV is inadequate due to study limitations.

Safety was a major consideration in developing engineering strategies for the new OVV that was described in this paper. VACV *thymidine kinase* (*VV-tk*) gene was deleted to enhance tumor selectivity to address concerns of uncontrolled replication in non-target tissues, as described in our other OVV projects [11,12], and herpes simplex virus (type 1) *thymidine kinase* (*HSV-tk*) transgene was inserted for viral replication control. The rationale supporting these strategies is, as follows: first, because of the high level of shared homology between the *VV-tk* gene and the human *thymidine kinase 1* (*TK1*) gene, VACV lacking a functional *tk* gene can efficiently replicate in cancer cells, which often express high levels of TK1, a human cytosolic enzyme which plays the primary role in regulating intracellular thymidine pools throughout the cell cycle. [13,14]; second, the *HSV-tk* gene is the most widely used suicide gene, which has been inserted into different virus backbones, mostly adenovirus [15,16,17], to be used in combination with antiviral drugs, such as ganciclovir (GCV) [18]. Such combination takes effect relying on the rate-limiting step that HSV-tk mediates in converting the prodrug GCV into the cytotoxic GCV-triphosphate: HSV-tk converts GCV into GCV monophosphate, the latter is subsequently converted to GCV-diphosphate and GCV-triphosphate by guanylate kinase and cellular kinase, respectively [19]. It is GCV-triphosphate that eventually halts the DNA replication of both the virus and host cancer cell. The *HSV-tk* transgene insertion might not only protect patients from uncontrolled viral replication, but also improve the new OVV’s anti-tumor efficacy. In our study, we focused on its suicidal effect on viral replication. HSV-tk and VV-tk belong to two different families of thymidine kinases (HSV-tk is type I and VV-tk is type II) and demonstrate distinct molecular weights, quaternary structure, and variable substrate specificity [20]. E.g., HSV-tk can catalyze both purine and pyrimidine analogues, while VV-tk can only catalyze pyrimidine analogues. This distinction cumulatively explains why HSV is sensitive to GCV, a purine analogue, while VACV is not [21]. However, the insertion of the wild-type *HSV-tk* gene into the *VV-tk* region of an OVV may raise concern for impaired tumor selectivity, while considering that the HSV-tk can also catalyze thymidine (a pyrimidine deoxynucleoside), since *VV-tk* gene deletion results in tumor selectivity [11,12,22]. Therefore, in this study, an artificial selection procedure while using bromodeoxyuridine (BrdU) was adopted to exclude any *HSV-tk* incorporated recombinant viruses expressing thymidine kinase activity. BrdU is an antipoxviral thymidine analogue [23], so virus strains showing high affinity to thymidine analogues were eliminated during the selection procedure [24]. The BrdU-selected candidate viruses should be further screened for tumor selectivity and *HSV-tk* transgene functionality as a GCV-mediated viral replication control.

Therefore, the primary objectives of this study were to engineer and characterize the new *HSV-tk* incorporated OVV in terms of tumor selectivity and enzymatic function (i.e., GCV sensitivity), and the secondary objective was to evaluate the influence of GCV combination on the anti-tumor effects and safety of the OVV in animal models.

## 2. Results

### 2.1. Newly Engineered OVV Expressing a Mutant HSV-tk Transgene

After homologous recombination, 120 luciferase and *HSV-tk* positive plaques were harvested for further screening (Figure A1A,B in Appendix A). The western blotting against HSV-tk that was conducted in the 120 candidate plaques showed that none of these plaques expressed a full-length 40.9 kDa HSV-tk; instead, these plaques consist of at least one of the three virus clones that express different truncated forms of HSV-tk: 36.1, 24.5, and 19.7 kDa, named OTS-412, OTS-C1, and OTS-C2, respectively (Figure 1A). Genetic sequencing demonstrated that OTS-412 has a C>T point mutation at the c.991 position, and OTS-C1 and OTS-C2 have a G-insertion and G-deletion mutation, respectively, at the c.430 position (Figure 1B). Among the three newly engineered viruses (OTS-412, OTS-C1, and OTS-C2), OTS-412 showed the highest sensitivity to GCV inhibition in NCI-H460 cells, and was, therefore, selected for further studies (Figure A1C).

### 2.2. Viral Replication in Human Cancer and Normal Cell Lines

OTS-412, OTS-C1, and OTS-C2 showed similar replication ability to that of the VV^tk-^ (control virus, a *VV-tk* deleted OVV) in the four human cancer cell lines tested (A-549, HCT 116, NCI-H460, and U2OS). All four viruses replicated rapidly in the first 24 h, and replication then slowly reached a plateau between 24 and 48 h; in addition, all four viruses showed the lowest replication ability in the NCI-H460 cell line (Figure 1C). OTS-412 showed similar or slightly lower replication (<10-fold difference) than Wyeth VACV in human cancer cell lines, and remarkably lower replication (>10 fold difference) in all the murine cancer cell lines, in the in vitro study that compared viral load of OTS-412 to Wyeth VACV while using qPCR (Figure A2).

In human normal cell line HUVEC-C, a high concentration (10 PFU/cell) of Wyeth VACV infection caused considerable cytopathic changes at 24 h post infection, but the same concentration of OTS-412 did not show significant cytopathic changes when compared to the negative control (Figure 1D). The luciferase expression of OTS-412 (10 PFU/cell) in HUVEC-C cells was 1/10 of that in HT-29 cancer cells after 24-h infection (Figure 1E).

### 2.3. OTS-412 Remained Stable over Serial Passages

The genetic sequences of the 1st passage and the 12th passage of purified OTS-412 remained identical, which differed from the wild-type *HSV-tk* gene at c.991 (Figure 2A). The mutations in OTS-C1 (c.430 8G) and OTS-C2 (c.430 6G) were not found in OTS-412 (Figure 2B). Additionally, the HindIII restriction enzyme digestion study showed that the 1st passage and 12th passage of purified OTS-412 had identical restriction mapping patterns, which differed from that of the Wyeth VACV (Figure 2C).

### 2.4. Tumor Specificity of OTS-412 in Mouse Models

A bioluminescence assay was conducted on day 2, post single-dose IT injection of either OTS-412 or VV^tk-^ in RenCa tumor-bearing mice on day 0 to examine the impact of the *HSV-tk* insertion on the virus level in tumor (Figure 3A); the luciferase signal levels were not statistically different between the two viruses (Figure 3B). To find out whether the *HSV-tk* transgene impacts the tumor selectivity of OTS-412, a bioluminescence assay was conducted on day 7, after single-dose IT injection of OTS-412 on day 0 in HCT 116 tumor-bearing mice; the luciferase signal was detected in tumors, but not in other organs (Figure 4A). In a Vx2 tumor bearing rabbit study, 5 weeks after two high-dose (on day 0 and 14, 5 × 10^8^ PFU) intravenous (IV) injections of OTS-412, fluorescence microscopy of VACV-specific antibody did not detect any signal in any normal issues other than tumor (Figure 4B).

### 2.5. Replication of OTS-412 Is Susceptible to GCV Inhibition in Mouse Models

A bioluminescence assay was conducted seven days after the single-dose IT injection of OTS-412 in an HCT 116 tumor-bearing mouse model, with or without GCV treatment, to confirm the effects of GCV treatment on viral replication. The luciferase signal in tumor was significantly higher in the OTS-412 group than in the OTS-412 + GCV group (Figure 5A). The viral copy number in the tumor tissue was also significantly higher (*p* = 0.004) in the OTS-412 group (7.0 × 10^7^/mg tumor) than in the OTS-412 + GCV group (3.2 × 10^7^/mg tumor) (Figure 5A). The tumors were isolated on sacrifice day (day 12). The immunofluorescence staining showed that the VACV-specific fluorescence signal was considerably more widespread and stronger in the tumor from the OTS-412 group than that from the OTS-412 + GCV group; in contrast, the OTS-412 + GCV group showed significantly fewer nuclei in the center of the tumor than the OTS-412 and control groups (Figure 5B).

### 2.6. Cytotoxicity of OTS-412 with or without GCV

In two human (Caki-1 and PC-3) and two murine (RenCa and 4T1) cancer cell lines, after 48 h, the cytotoxicity of OTS-412 was similar to that of VV^tk-^; both viruses showed considerably stronger cytotoxicity in human cancer cells than in murine cancer cells (Figure 6A). The IC_50_ (half maximal inhibitory concentration) of OTS-412 among 12 human cancer cell lines showed that half of the tested cancer cell lines (SK-MEL-2, SK-MEL-5, A-549, NCI-H23, DU145, and U-87MG) were relatively sensitive to OTS-412 (IC_50_ < 0.1 PFU/cell), and NCI-H460 was relatively resistant to OTS-412 (IC_50_ > 1 PFU/cell) (Figure 6B). In addition, following the OTS-412 cytotoxicity tests, the live-cell imaging assay showed that combination with GCV increased cancer cell apoptosis in all the three representative cancer cell lines: sensitive (A-549), moderate (HCT 116), and resistant (NCI-H460), when compared to OTS-412 alone; signals were not quantified (Figure 6C).

### 2.7. Anti-Tumor Effects and Safety of OTS-412 with or without GCV

In the HCT 116 tumor-bearing mice, single-dose IT injection of OTS-412 resulted in significant tumor reduction, which started around seven days after OTS-412 injection; on sacrifice day (day 12), the tumor volume in the control group (580 mm^3^) was significantly higher than that of the OTS-412 group (200 mm^3^, *p* = 0.003) and the OTS-412 + GCV group (160 mm^3^, *p* < 0.001) (Figure 7A). The immunofluorescence assay of the tumor tissue showed higher levels of tumor apoptosis in the OTS-412 + GCV combination group than in the OTS-412 monotherapy group, and the lowest apoptosis in the control group, but the differences were not statistically significant (Figure 7B). Meanwhile, multiple-dose IP injection of OTS-412 also showed considerable anti-tumor effects, but less than that of the IT injection; on sacrifice day (day 21), the tumor volume was the largest in the control group (770 mm^3^), followed by the OTS-412 group (490 mm^3^) and OTS-412 + GCV group (330 mm^3^), but the differences were not statistically significant (Figure 7C). The mouse body weight remained within the normal range (±20%) in the different treatment groups of both studies (Figure A3).

## 3. Discussion

In this study, OTS-412, an OVV containing a mutant herpes simplex virus (type-1) *HSV-tk* transgene was engineered and then characterized. The mutant *HSV-tk* transgene encodes a truncated (36.1 kDa) HSV-tk; both the transgene and protein product demonstrate stability over serial passages. Functionally, the truncated HSV-tk remains sensitive to GCV, which is key in controlling viral replication. Notably, the mutant *HSV-tk* transgene insertion does not influence tumor selectivity and cytotoxicity of OTS-412, as compared to that of VV^tk-^.

In this study, three recombinant viruses were rescued by BrdU selection: OTS-412, OTS-C1, and OTS-C2. Each of the three viruses has a different mutant *HSV-tk* transgene (Figure 1B). OTS-C1 and OTS-C2 express truncated HSV-tk due to a G-insertion and a G-deletion, respectively, in the 7G homopolymer region of the *HSV-tk* transgene (c.430 7G > 8G/6G). The 8G and 6G mutations were previously identified in HSV (type 1) strains that were isolated from a patient who demonstrated resistance to GCV treatment [25]. Because both the insertion and deletion cause frameshifts that result in an amino acid change from position 146, which covers the entire HSV-tk nucleoside binding domain (NBD), the mutant HSV strains completely lost GCV sensitivity [25]. However, the point mutation in the *HSV-tk* (c.991C>T) transgene from the OTS-412 strain does not cause a frameshift, but instead, a 46-residue truncation (330 residues remained) at the C-terminus. The intact NBD might explain why OTS-412, unlike OTS-C1 and OTS-C2, is still sensitive to GCV inhibition. Although a previous study showed that the C-terminus is important in maintaining acyclovir (ACV) phosphorylation activity [26], our study shows that GCV sensitivity is not affected by the 46-residue truncation at the HSV-tk C-terminus. This is in agreement with a previous study [27] and shows that *HSV-tk* mutations affect purine analogues differently. Meanwhile, this point mutation in the *HSV-tk* (c.991C>T) transgene is likely to have caused the remarkable decrease in thymidine affinity, which allowed OTS-412 to successfully survive the BrdU selection. BrdU and GCV are pyrimidine and purine analogues, respectively. A previous study shows that the wild-type HSV-tk has higher affinity for pyrimidine analogues than for purine analogues [28], but *HSV-tk* genetic mutations in the NBD can affect the enzyme’s pyrimidine and purine affinity: e.g., an amino acid residue change at position 167 from alanine to tyrosine (A167Y) can selectively abolish the HSV-tk’s pyrimidine affinity completely, while showing less of an impact on purine affinity [28]. Additionally, several amino acid residue changes within positions 158 to 174 do not only impair the HSV-tk’s pyrimidine affinity, but can also actually increase its purine affinity [29,30]. Although this current study will not investigate how the point mutation in the *HSV-tk* (c.991C > T) transgene might have changed the enzyme’s thymidine and purine affinity, future studies may help to elucidate the mechanism behind this change.

The three newly engineered viruses showed similar viral replication ability as compared to VV^tk-^ in all the cell lines tested. Notably, all three virus strains showed the highest and lowest replication in the A-549 and NCI-H460 cancer cell lines, respectively. This is consistent with a previous study that showed that the replication of GLV-1h68 (a *VV*-*tk* gene deleted OVV) in A-549 cells was one of the highest among the 74 cancer cell lines tested; in contrast, the replication of GLV-1h68 in the NCI-H460 cells was at the lower end of the cell panel [31]. The replication of Wyeth VACV was higher than OTS-412, as we expected; however, the Wyeth VACV also showed the lowest replication level in the NCI-H460 cells when compared to all of the other human cancer cell lines tested. This pattern suggests that the cellular TK1 level might not be the main determinant of VACV and OVV replication for all cancer cell lines. It is known that other mechanisms, such as extracellular signal-regulated kinase (ERK) [32] and ribonucleotide reductase (RR) [33] expression, can also influence viral replication in cancer cells. Consequently, different double-gene deletion strategies have been applied to further improve the tumor selectivity of OVV [8,22].

Keeping in mind that mutations in transgenes are not uncommon during recombinant virus engineering [34,35]. Stability tests are often required by regulatory authorities (e.g., US FDA and EMA) to monitor transgenes that are integrated into cell/gene therapy products. The general recommendation is that inserted transgenes should be tested from both the master virus bank (MVB) and the working virus bank (WVB), which usually covers 3–5 successive passages. The *HSV-tk* inserted VACV was first reported 30 years ago [36], but the stability of the *HSV-tk* transgene in foreign virus backbones has not been previously established in the literature. In the current study, the stability of the mutant *HSV-tk* transgene in OTS-412 was confirmed by restriction enzyme digestion and gene sequencing after more than 10 serial passages, and the OTS-412 used in the animal studies were the 4th–6th passages (WVB) produced from the master virus seed (MVS) of OTS-412 (1st passage).

This study shows that, even though OTS-412 incorporated a mutant *HSV-tk* gene into its *VV-tk* gene region, it has preserved the tumor selective characteristic of a *tk* gene deleted OVV. In addition, the levels of OTS-412 in tumor tissue were similar to that of VV^tk-^, suggesting that the in vivo tumor replication of the two viruses is comparable.

The co-administration of GCV significantly reduced over half of the OTS-412 replication in tumor tissue in the HCT 116 human colon cancer bearing mouse model. The immunofluorescence assay clearly showed that, with the GCV co-administration, the OTS-412 replication was restricted to the edge of necrotic tumor tissue and the signal was nearly absent in the center of the tumor. Together with the Incucyte live-cell imaging assay, these results suggest that GCV combination leads to an increase in the cytotoxicity of OTS-412, despite reduced OTS-412 replication. GCV enhanced cytotoxicity will be addressed later on in the mouse study.

The in vitro tumor cytotoxicity tests showed that: (1) OTS-412 has comparable cytotoxicity to VV^tk-^ in both human and murine cancer cells; however, the murine cancer cell lines are generally resistant to OVV; and, (2) OTS-412 generated potent cytotoxicity (IC_50_ < 0.1 PFU/cell) in half of (six out of twelve) the human cancer cell lines that were tested. In contrast, the NCI-H460 lung cancer cell line was relatively resistant (IC_50_ > 10 PFU/cell) to OTS-412. The Incucyte live-cell imaging assay that was conducted in three representative cell lines (A-549 for sensitive, HCT 116 for moderate, and NCI-H460 for resistant cell line to OTS-412) further confirmed the cytotoxicity tests. As discussed earlier, the mechanisms underlying the susceptibility or resistance of cancer cells to OTS-412 replication and cytotoxicity may involve other factors besides TK1 expression level. Furthermore, there might be some general determinants for cancer cells’ sensitivity to most, if not all, oncolytic viruses. This is suggested by the consistent sensitivity patterns amongst the three cancer cell lines tested when treated with OTS-412 and other oncolytic viruses [32,37,38]. However, there are exceptions, e.g., HT-29 cells seem to be much more sensitive to VACV than to other viruses, such as adenovirus and alphavirus [32,37,38]. In future oncolytic virus studies, it will be imperative and valuable to identify the biomarkers for predicting cancer sensitivity or resistance to oncolytic viruses.

The xenograft mouse model of HCT 116 human colon cancer, a cancer with moderate sensitivity to OTS-412 in vitro, was chosen for the testing of OTS-412 anti-tumor effects in vivo. Single IT injection of OTS-412 led to faster tumor regression when compared to multiple IP injections (two times/week, five times). This is not surprising, because systemic administration elicits a much more potent innate immune reaction, which will reduce the amount of virus that reaches the tumor [39]. In addition, the co-administration of GCV led to slower tumor growth in the multiple-dose IP study, although the difference was not statistically significant. No such tendency was seen in the single-dose IT study, which is probably because the study duration was too short (all mice were sacrificed on day 12); however, immunofluorescence assay for the single IT study indicated more tumor necrosis in the GCV co-administration group than in the OTS-412 only group, which suggested more potent cytotoxicity was induced by the suicidal effects of HSV-tk/GCV. However, in subsequent animal studies, the antitumor effects of OVV treatment alone did not amount to any survival benefit (Figure A4). Various attempts were made, such as combining OTS-412 with immune modulators (e.g., myeloid cell inhibitors) and/or immune checkpoint inhibitors (e.g., PD-L1 inhibitors), and optimizing OTS-412 dosing schedule (e.g., step-up dosing), to improve the treatment outcomes for OTS-412. The efficacy of these optimized OTS-412 treatments will be reported in our following manuscripts.

This study has several limitations. (1) The number of animals in each study was small due to the fact that the overall study scale was large, which might have prohibited us from drawing statistical conclusions, even though a trend is observed. (2) Although the loss of pyrimidine catalyzing activity in OTS-412 was indirectly demonstrated through attenuated viral replication in various cancer cell lines, and through tumor selectively in animal models, we did not directly show the loss of function of the phosphorylation of thymidine (e.g., through an in vitro enzyme assay). (3) We did not clarify the molecular mechanism that underlies “HSV-tk truncation losing thymidine sensitivity without affecting purine sensitivity”.

## 4. Materials and Methods

### 4.1. Cell Lines and Virus

HUVEC-C, 143B, A-549, HeLa, U2OS, and 4T1 cell lines were purchased from the American Type Culture Collection (ATCC, Manassas, VA, USA). Caki-1, DU145, HT-29, HCT 116, MCF-7, NCI-H23, NCI-H522, NCI-H460, PC-3, SK-MEL-2, SK-MEL-5, SK-MEL-28, U-87MG, CT-26, and RenCa were obtained from the Korean Cell Line Bank (KCLB, Seoul, Korea). The cells were maintained with ATCC and KCLB recommended media, respectively, and supplemented with 10% fetal bovine serum (FBS). The Wyeth-calf adapted strain VACV (VR-1536, New York City Department of Health Laboratories) was purchased from the ATCC, amplified in HeLa cells, and quantified while using a VACV titration protocol [40]. In this study, all of the incubation and infection steps were performed at 37 °C in 5% CO_2_, and all chemicals were purchased from Sigma-Aldrich (Merck KGaA, Darmstadt, Germany), unless otherwise specified.

### 4.2. Virus Engineering

Two genes, the wild-type *HSV-tk* gene (NCBI GenBank: J02224.1) and firefly luciferase gene (Addgene sequence: 12178), were placed under the control of pSE/L (vaccinia synthetic early/late promotor) and p7.5 (vaccinia early/late promotor), respectively [41] and flanked by the *VV-tk* gene NCBI GenBank: AY243312.1/VACWR094) left-end (496bp) and right-end (400 bp). In addition, early transcriptional termination consensus sequence (5′-TTTTTAT-3′) was added to the 3′ end of the oligonucleotide. The recombinant DNA was synthesized and confirmed by sequencing, and then cloned into a pUC57 plasmid; the shuttle plasmid was named pOTS. Meanwhile, a control plasmid was constructed by removing the *HSV-tk* cassette while using PmeI-HindIII digestion, followed by blunt end ligation. Genewiz Inc. synthesized all of the shuttle plasmids (South Plainfield, NJ, USA). Next, homologous recombination was conducted to transfer the transgenes into the wild-type vaccinia backbone; HeLa cells were cultured overnight in six-well plates up to 90% confluence and then infected with the Wyeth VACV at 0.05 PFU/cell for 2 h. The engineered shuttle plasmids were linearized by SpeI-HF restriction enzyme (R3133L, NEB, Ipswich, MA, USA) and then mixed with Xfect transfection reagent (631317, Clontech, Mountain View, CA, USA). The mixture was incubated at room temperature for 10 min. and then transfected into the aforementioned Wyeth VACV-infected HeLa cells. Finally, the reaction mixture was incubated in Dulbecco’s modified eagle medium (DMEM), supplemented with 2% FBS for 72 h. The homologous recombination was confirmed by a firefly luciferase assay (E1500, Promega, Madison, WI, USA), according to the manufacturer’s protocol. The firefly luciferase signal-positive viruses were then harvested for screening. With a similar method, as described above, the *VV-tk* gene deleted/*Fluc* gene inserted virus, VV^tk-^, was engineered as a control virus.

### 4.3. Screening of Engineered Viruses

The recombinant viruses were further cultured in the 143B (*TK*-) cell line with 10% Eagle’s minimum essential medium (EMEM) media containing 15 µg/mL BrdU for 72 h. BrdU is a thymidine analogue and was used in this study to create selection pressure against viruses expressing functional thymidine kinase [24]. >100 plaques were isolated after multiple passages of BrdU selection, and the *HSV-tk* transgene insertion was confirmed using western blotting against HSV-tk as follow:

Hela cells were cultured (4 × 10^5^ cells/well) in six-well plates overnight and infected with candidate plaques at 10 μL/cell for 24 h. The virus-infected cells were lysed while using radioimmunoprecipitation assay (RIPA) lysis buffer (R4100-010, GenDEPOT, Fort Worth, TX, USA), and then centrifuged at 14,000× *g* for 10 min. at 4 °C. The supernatants were loaded for gradient (4–20%) sodium dodecyl sulfate polyacrylamide gel electrophoresis (SDS-PAGE) and then transferred to polyvinylidene pluoride (PVDF) membranes (IPVH00010, Millipore, Burlington, MA, USA). The membranes were incubated overnight at 4 °C with primary antibody against HSV-tk (SC-28037, Santa Cruz, CA, USA) and then washed three times with PBS containing 0.1% Tween 20. Next, the membranes were incubated with the secondary antibody, a goat IgG-horseradish peroxidase-conjugated antibody (A50-101P, Bethyl Laboratories, Montgomery, TX, USA), at room temperature for 1 h. Protein conjugates were then detected while using a chemiluminescent imaging system (CAS-400SM, Davinch-K, Seoul, Korea).

The western blotting showed that all of the BrdU selected plaques were a mix of at least one of three viruses that express distinct truncated HSV-tk: 36.1, 24.5, and 19.7 kDa (full-length HSV-tk: 40.9 kDa), which were named OTS-412, OTS-C1, and OTS-C2, respectively. The densities of the blots were not quantified because the western blotting was conducted to confirm whether or not the transgene HSV-tk was successfully inserted into the Wyeth VACV backbone. A further selection process was done based on the three candidate viruses’ GCV sensitivity, and OTS-412 was selected as the final virus candidate.

### 4.4. Measurement of Viral Replication In Vitro

Four human cancer cell lines (A-549, U2OS, HCT 116, and NCI-H460) were cultured in a 24-well (1 × 10^5^ cells/well) plate overnight and then infected with one of following viruses (0.1 PFU/cell): Wyeth VACV, OTS-412, OTS-C1, or OTS-C2. The samples were harvested at 0, 24, 48, and 72 h post infection, and titers were determined while using plaque assays.

For plaque assay, the U-2 OS cells were cultured in six-well (5 × 10^5^ cells/well) plates overnight and then infected with serially diluted virus stock for 2 h. The infection medium was then replaced by 1.5% carboxymethylcellulose and 2% FBS supplemented medium. The samples were incubated at 37 °C in 5% CO_2_ for 72 h, and then stained with 0.5% crystal violet in 20% ethanol. Viral titer was calculated while using the formula: Viral titer (PFU/mL) = Average plaque number/Dilution factor × Volume (mL).

HUVEC-C cells were seeded in 96 well plates at 3 × 10^4^ cells/well and then infected with Wyeth VACV or OTS-412 (10 PFU/cell) to evaluate the cytopathic effect of OTS-412 in the normal human cell line. After 24-h incubation, a phase-contrast photomicrographs were taken while using Incucyte S3 live-cell imaging system (Essen BioScience, Ann Arbor, MI, USA).

HUVEC-C and HT-29 cells were seeded in 96 well plates at 3 × 10^4^ cells/well and infected with OTS-412 (10 PFU/cell) to compare the viral replication of OTS-412 in human normal cells and cancer cells. After 24-h infection, 20 μL supernatant was aspirated for the measurement of luciferase expression while using a luciferase assay kit (E1500, Promega, Madison, WI, USA).

### 4.5. Gene Sequencing

Gene sequencing was conducted to compare the sequence of the wild-type *HSV-tk* gene to the mutant *HSV-tk* transgenes from OTS-C1 and OTS-C2, as well as the sequences that were obtained from the 1st and 12th passage of OTS-412.

A-549 cells were cultured in six-well plates (4 × 10^5^ cells/well) overnight and then infected with the engineered viruses at 0.1 PFU/cell for 24 h. The total DNA (cell and virus) was extracted using a DNA extraction kit (69504, Qiagen, Valencia, CA, USA) and the DNA concentration was determined while using spectrophotometry (Nanodrop2000, Thermo Fisher Scientific, Waltham, MA, USA). The *HSV-tk* transgene was amplified while using a HelixAmp™ Ready-2x-Go polymerase chain reaction (PCR) kit (PMD008L, Nanohelix, Daejeon, Korea) with the following condition: initial denaturation at 95 °C for 2 min; followed by 30 cycles of denaturation at 95 °C for 45 s, 58 °C for 45 s, 72 °C for 90 s; extension at 72 °C for 5 min. The forward and reverse primers used for the *HSV-tk* transgene amplification were 5′-CCT CGT CGC AAT ATC GCA TTT T-3′ and 5′-CTC CAG CGG TTC CAT CTT C-3′. The PCR products were analyzed by gene sequencing (Cosmogenetech Inc, Seoul, Korea). The internal sequencing primers that were used for this analysis were 5′-AGT TAG CCT CCC CCA TCT CC-3′, 5′-CGA CAG ATC TAG GCC TGG TA-3′, 5′-CCC TGC TGC AAC TTA CCT CC-3′, and 5′-CTC CAG CGG TTC CAT CTT C-3′, respectively.

### 4.6. Restriction Enzyme Digestion

A HindIII restriction enzyme digestion was used to compare the fragment patterns of OTS-412 from the 1st passage and the 12th passage to the Wyeth VACV. The U-2 OS cells were infected with the Wyeth VACV or OTS-412 at 0.1 PFU/cell and then incubated for 48 h. Viral DNA extraction was performed while using a method previously reported [42]. The nuclei of U-2 OS cells were removed after the cells were treated with cytoplasmic lysis buffer in order to obtain virus stocks (10 mM Tris-HCl [pH8.0], 10 mM KCl, 5 mM Na_2_EDTA). The collected virus stocks were homogenized using a 1 mL syringe and then treated with lysis buffer (54% sucrose, 2-mercaptoethanol, proteinase-K, 10% SDS, and 5M NaCl). Finally, the viral DNA was extracted from each virus lysate sample (QIAquick Gel Extraction Kit, Qiagen, Hilden, Germany), and the DNA concentration was quantified (NanoQuant Plate, Tecan, Männedorf, Swiss). For each virus, 2.5 μg DNA was digested with HindIII-HF restriction enzyme (R3104S, NEB, Ipswich, MA, USA) at 37 °C in water bath for 12 h. The digested DNA was run on a 0.6% agarose gel at 50 V for 220 min. Images were captured while using a chemiluminescent imaging system (CAS-400SM, Davinch-K, Seoul, Korea).

### 4.7. Cytotoxicity Analysis In Vitro

For the cell viability assay, 12 human cancer cell lines (SK-MEL-2, SK-MEL-5, SK-MEL-28, A-549, NCI-H23, NCI-H533, NCI-H460, DU145, PC-3, HT-29, HCT 116, and U-87 MG) were seeded in 96-well plates (3000 cells/well), incubated overnight, and then infected with OTS-412 at various doses (0.1–1 PFU/cell) for 72 h. The cell viability assay was then performed using a cell counting kit (CCK-8, Dojindo, Kumamoto, Japan), and quantified using a microplate spectrophotometer (Spark, Tecan, Männedorf, Swiss) at 450 nm. IC_50_ was calculated while using Prism version 8 (GraphPad Software, La Jolla, CA, USA).

For live-cell imaging analysis, A-549, HCT 116, and NCI-H460 cells were seeded in 96-well plates at 1 × 10^4^ cells/well, incubated overnight, and then infected with OTS-412 (0.1 PFU/cell) with or without GCV (50 uM). After 48-h incubation, the cells were stained with the Invitrogen LIVE/DEAD^®^ Cell Imaging kit (R37601, Invitrogen, Carlsbad, CA, USA), according to the manufacturer’s protocol. The wells were then scanned by Incucyte S3 (Essen BioScience, Ann Arbor, MI, USA) to detect live and dead cells. The GCV concentration that was used here (50 μM) is similar to human plasma *C*_max_ (maximum concentration) following an IV injection of GCV of 5 mg/kg, which is around 10 μg/mL [43].

### 4.8. Evaluation of OTS-412 in Animal Models

Two separate mouse studies, single intratumoral (IT) and multiple intraperitoneal (IP) injections of OTS-412, were conducted while using a HCT 116 xenograft mouse model. In both studies, female BALB/c nude mice (Orient Bio, Seongnam, Korea) were subcutaneously injected with HCT 116 cells on the right flank and developed tumors. Once the tumors reached a volume of around 50–200 mm^3^, the mice were stratified into different treatment groups, each group consisted five mice.

In the first experiment (single-dose study, noted as Study STG), each mouse was treated with either OTS-412 (single-dose, day 0, IT, 1 × 10^6^ PFU), or OTS-412 (single-dose, day 0, IT, 1 × 10^6^ PFU) + GCV (once/day, day -3 to 11, IP, 25 mg/kg), or control (saline), and all of the mice were sacrificed on day 12. In the second experiment (multiple-dose study, noted as Study MVG), each mouse was treated with OTS-412 (five times: on day 0, 3, 7, 10, and 14, IP, 1 × 10^8^ PFU), or OTS-412 (five times: on day 0, 3, 7, 10, and 14, IP, 1 × 10^8^ PFU) + GCV (once/day, day 0 to 20, IP, 50 mg/kg), or control (saline), and all mice were sacrificed on day 20. Tumor volume and body weight data were recorded for all groups throughout the study. Tumor volume was calculated while using the equation: tumor volume (mm^3^) = 0.5 × longest diameter × shortest diameter^2^. The tumors were harvested for assays, as described in the following sections.

The virus dosages used in this study (10^6^ PFU for IT and 10^8^ PFU for IP) were based on our previous studies on other OVVs [11,12]. The GCV dosages that were used in this study (25 and 50 mg/kg/day) were commonly adopted by many other animal studies [44,45] and, according to a widely used human-animal drug dose translation equation [46], mouse dose of 50 mg/kg/day is equivalent to the recommended maintenance dosages for humans, which is around 5 mg/kg/day.

All of the mice were acclimatized to the animal facility for 3–4 days before the experiments, and all of the animal studies were in compliance with Institutional Animal Care and Use Committee (IACUC) guidelines from the Ministry of Food and Drug Safety. The Institutional Animal Care and Use Committee of Pusan National University, Busan, Korea (PNU-2016-1312) approved the animal study.

Other animal studies for tumor selectivity and tumor viral replication will be briefly described in the results section and in the figure legends. Pusan National University approves all of the animal studies within one project. Therefore, there was only one study number.

### 4.9. Immunofluorescence and TUNEL Assay

The tumor tissues were harvested and fixed in 10% neutral buffered formalin, embedded in paraffin, and sectioned at a thickness of 2 μm. After deparaffinization, tumor sections were co-stained with anti-VACV antibody (ab35219, Abcam, Cambridge, UK) and terminal deoxynucleotidyl transferase dUTP nick end labeling (TUNEL, C10618, Invitrogen, Waltham, MA, USA) to identify OTS-412-specific apoptosis. In addition, the sections were stained with Alexa Fluor 594 picolyl azide dye and anti-rabbit antibody combined with Alexa Fluor 488 (A11070, Invitrogen). 4′,6-Diamidino-2-phenylindole (DAPI, D9542, Sigma) was used for nuclear counterstaining. All of the tissue specimens were mounted with fluoromount medium (ADI-950-260-0025, Enzo Life Sciences, Ann Arbor, MI, USA). The images were acquired by inverted fluorescence microscopy (Eclipse Ti2, Nikon, Tokyo, Japan).

### 4.10. Quantification of Virus by qPCR

Total DNA was extracted and purified while using a QIAamp MinElute Virus Spin Kit (57704, QIAgen, Valencia, CA, USA) and the quantification of viral DNA copies was performed using TaqMan Universal PCR Master Mix (4304437, Applied Biosystems, Foster City, CA, USA) and QuantStudio 5 Real-Time PCR system (Thermo Fisher Scientific, Waltham, MA, USA), targeting the vaccinia-specific *E9L* gene (which encodes vaccinia viral DNA polymerase). The forward and reverse primers for the *E9L* gene were 5′-CAA CTC TTA GCC GAA GCG TAT GAG-3′ and 5′-GAA CAT TTT TGG CAG AGA GAG CC-3′, respectively. The probe was 5′-6-FAM-CAG GCT ACC AGT TCA A-MGB/NFQ-3′. The *E9L* gene was amplified under the following conditions: pre-denaturation at 50 °C for 2 min., denaturation at 95 °C for 10 min., followed by 40 cycles of denaturation at 95 °C for 15 s, and annealing/extension at 60 °C for 90 s. Meanwhile, a serial dilution (from 1 × 10^7^ to 2.5 copies/5 μL) of plasmid containing the *E9L* gene was used to generate a standard curve.

### 4.11. Bioluminescence Assay

The animals were anesthetized with N_2_O:O_2_ (7:3) mixed with isoflurane during scanning. Images were captured while using the Optix MX3 imaging system and analyzed using the Optix OptiView software (ART Advanced Research Technologies Inc, Montreal, QC, Canada). RenCa tumor-bearing syngeneic BALB/c mice were scanned on day 2 post single-dose IT injection of saline, VV^tk-^, or OTS-412. In two separate studies, HCT 116 human colon cancer bearing BALB/c nude mice were scanned on day 7 post a single-dose IT injection of OTS-412 at a dosage of 1 × 10^6^ PFU (Study STG, with or without GCV) or 1 × 10^7^ PFU.

### 4.12. Statistical Analysis

The *in vitro* experiments were conducted in duplicate or triplicate. All of the statistical analyses were performed while using the Prism 8 (GraphPad, La Jolla, CA, USA). Comparisons between groups were determined using t-test or ANOVA test. IC_50_ was calculated using nonlinear regression (the Dose-response-Inhibition module). *p* < 0.05 was considered to be statistically significant.

## 5. Conclusions

This current study was conducted to engineer a replication-controllable OVV and understand some of its basic characteristics. It shows that, due to the insertion of a mutant *HSV-tk* transgene in place of the *VV-tk* gene, the new OVV, named OTS-412, lacks affinity to pyrimidine analogues, but it remains sensitive to GCV, a purine analogue. These characteristics allow OTS-412 to replicate selectively in tumor cells, which often express high levels of TK1, and also allow for the use of GCV as a safety solution for potential uncontrolled viral replication. Furthermore, the mutant *HSV-tk* in OTS-412, as well as the truncated protein product, remain stable beyond serial passages. OTS-412 shows preliminary anti-tumor effects both in vitro and in vivo, and a combination of GCV does not influence the anti-tumor effects of OTS-412. The HSV-tk/GCV combination shows potential as a replication-control safety mechanism for future clinical applications of OTS-412.

## 6. Patent

OTS-412 was filed for the PCT application (2018-016874, priority data 28 December 2018); the patent is pending.

## Figures and Tables

**Figure 1 cancers-12-00228-f001:**
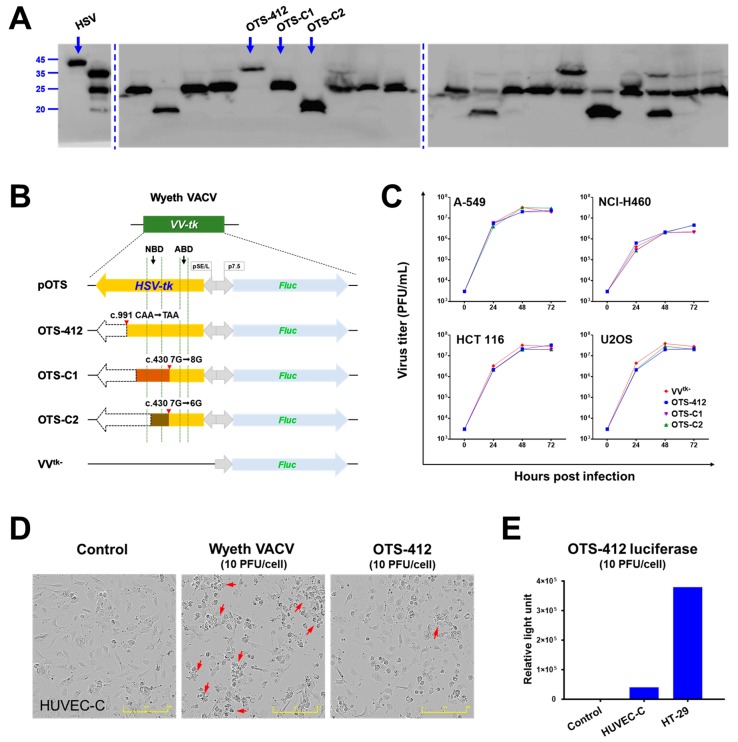
Comparison of three candidate viruses (OTS-412, OTS-C1, and OTS-C2) which express different truncated herpes simplex virus (type 1) thymidine kinase (HSV-tk). (**A**) Western blot of 20 representative plaques (from a total of 120 plaques) obtained after bromodeoxyuridine (BrdU) selection that did not express the full-length HSV-tk (far left lane), and instead consist of at least one of three different virus strains (OTS-412, OTS-C1, or OTS-C2), each expressing a different length of HSV-tk. Full-length gel images were shown in Figure A1. (**B**) *HSV-tk* and *Fluc* transgene cassettes in the vaccinia virus *thymidine kinase* (*VV-tk*) region of the three candidate viruses (flank regions not shown). *VV-tk* deleted OVV (VV^tk-^), the shuttle plasmid (pOTS), shadow regions corresponding to ATP-binding domain (ABD) and nucleoside-binding domain (NBD) of the HSV-tk molecule are included for reference. (**C**) Comparison of viral replication at 0, 24, 48, and 72 h between three virus strains (VV^tk-^, OTS-412, OTS-C1, and OTS-C2; at 0.1 PFU/cell) in four different human cancer cell lines, measured by plaque assay. (**D**) Cytopathic effects in human normal cell line HUVEC-C after 24-h infection with Wyeth vaccinia virus (VACV) or OTS-412 at 10 PFU/cell. Control is no viral infection. Arrows indicate cells showing cytopathic changes. Scale bar: 400 μm. (**E**) OTS-412 luciferase detected in HUVEC-C and HT-29 cells, after 24-h infection with OTS-412 at 10 PFU/cell.

**Figure 2 cancers-12-00228-f002:**
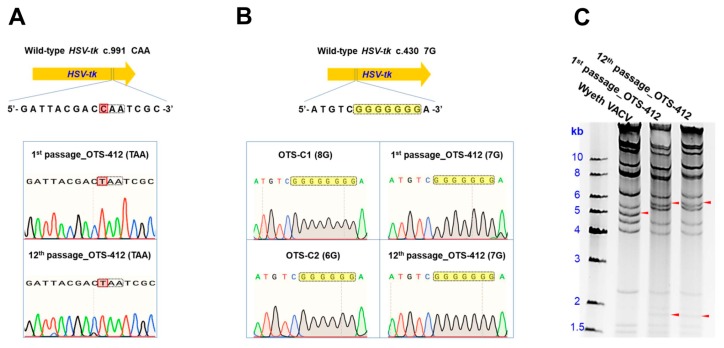
Establishment of genetic identity and stability of OTS-412. (**A**) Gene sequences around the point mutation (c.991 C>T) in the herpes simplex virus *thymidine kinase (HSV-tk*) transgene from the 1st passage and the 12th passage of OTS-412. (**B**) Gene sequences around the G-insertion/G-deletion mutation (c.430 7G) of the *HSV-tk* transgene from OTS-C1/ OTS-C2 compared to the 1st passage and the 12th passage of OTS-412. (**C**) HindIII restriction enzyme digestion patterns from Wyeth vaccinia virus (VACV), the 1st passage and the 12th passage of OTS-412. The arrows indicate unmatched fragments between the Wyeth VACV and OTS-412.

**Figure 3 cancers-12-00228-f003:**
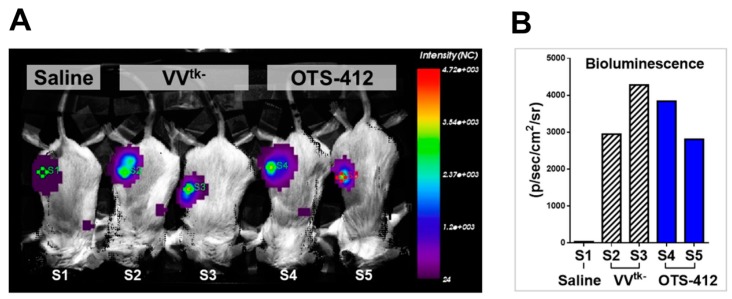
Viral replication of OTS-412 in tumor-bearing mouse models. (**A**) Tumor scan using bioluminescence assay conducted two days post single-dose intratumoral (IT) injection of saline (S1), *VV-tk* deleted oncolytic vaccinia virus (VV^tk-^) (1 × 10^7^ PFU, S2, and S3), and OTS-412 (1 × 10^7^ PFU, S4, and S5) in syngeneic RenCa tumor-bearing mice. Green diamonds (S1–S4) and red diamond (S5) in the tumor center are indicators of region of interest. (**B**) Quantification of the luciferase signal showing no statistical difference between the VV^tk-^ and OTS-412 treated mice.

**Figure 4 cancers-12-00228-f004:**
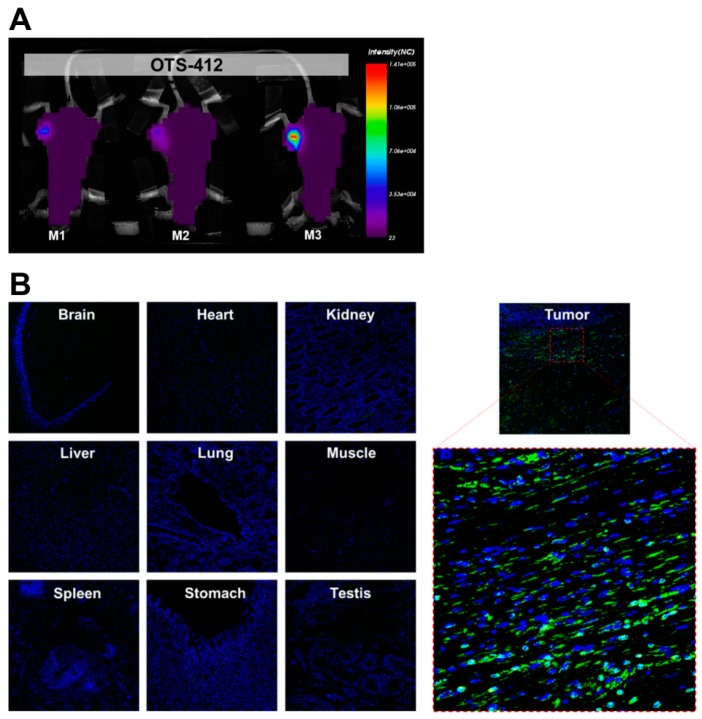
Tumor selectivity of OTS-412 was demonstrated by (**A**) a whole body bioluminescence scan conducted seven days post a single-dose intratumoral (IT) injection of OTS-412 (1 × 10^7^ PFU) in HCT 116 tumor-bearing mice (M1, M2, and M3) showed luciferase signal in tumor tissue only; and (**B**) merged confocal microscopy of immunohistochemical staining of multiple organ tissues and tumor harvested from a Vx2 tumor bearing rabbit, five weeks after two separate intravenous (IV) injections of OTS-412 (on day 0 and 14, 5 × 10^8^ PFU). Green signal is from vaccinia virus (VACV)-specific antibody A27L representing OTS-412 infected cells, and blue signal is from cell nuclei. Images are shown at ×100 magnification, the framed area is shown at ×400 magnification.

**Figure 5 cancers-12-00228-f005:**
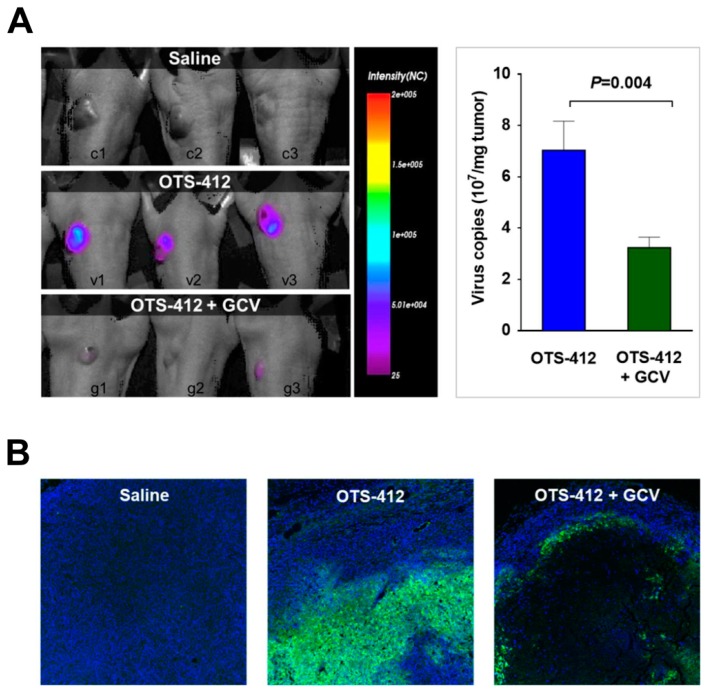
Inhibition of OTS-412 replication by ganciclovir (GCV) in vivo. (**A**) (Study STG) Bioluminescence assay of viral replication in HCT 116 tumor tissue on day 7, post a single-dose injection of saline, OTS-412 (day 0, intratumoral [IT], 1 × 10^6^ PFU), or OTS-412 + GCV (day 0, IT, 1 × 10^6^ PFU + once/day, day -3 to 11, intraperitoneal [IP], 25 mg/kg, respectively). The histogram shows qPCR quantification of virus copy number in tumor tissue harvested on day 12 from the abovementioned study. (**B**) (Study MVG) Immunofluorescence assay shows viral replication in HCT 116 tumor tissue on day 21, after multiple doses of saline, OTS-412 (on day 0, 3, 7, 10, and 14, IP, 1 × 10^8^ PFU), or OTS-412 + GCV (0, 3, 7, 10, and 14, IP, 1 × 10^8^ PFU + once/day, day 0 to 20, IP, 50 mg/kg/mice, respectively). Green staining is from vaccinia virus (VACV) specific antibody (A27L) representing OTS-412 infected cells, blue staining is from cell nuclei. Images are shown at ×100 magnification.

**Figure 6 cancers-12-00228-f006:**
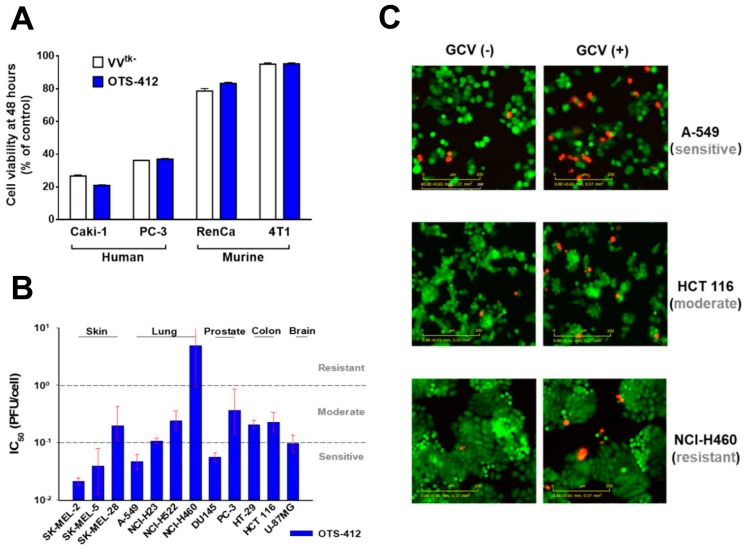
Cytotoxicity of OTS-412 in cancer cell lines. (**A**) Comparison of 48-h cytotoxicity of *VV-tk* deleted oncolytic vaccinia virus (VV^tk-^) (1 PFU/cell) and OTS-412 (1 PFU/cell) in human (Caki-1 and PC-3) and murine (RenCa and 4T1) cancer cell lines. (**B**) IC_50_ of 48-h cytotoxicity of OTS-412 in 12 human cancer cell lines; the dash line indicates a suggested division between OTS-412 sensitive (IC_50_ < 0.1 PFU/cell), moderate (1 > IC_50_ ≥ 0.1 PFU/cell), and resistant (IC_50_ ≥ 1 PFU/cell) cell lines. The error bars represent 95% confidence intervals of the IC_50_, the upper confidence limit for NCI-H460 is infinity. (**C**) Comparison of 48-h cytotoxicity of OTS-412 (0.1 PFU/cell), with or without ganciclovir (GCV) (50 μM) combination, in three human cell lines (A-549, HCT 116, and NCI-H460) by Incucyte live-cell imaging assay. Red signal was from dead cells and green signal was from viable cancer cells. Scale bar: 200 µm.

**Figure 7 cancers-12-00228-f007:**
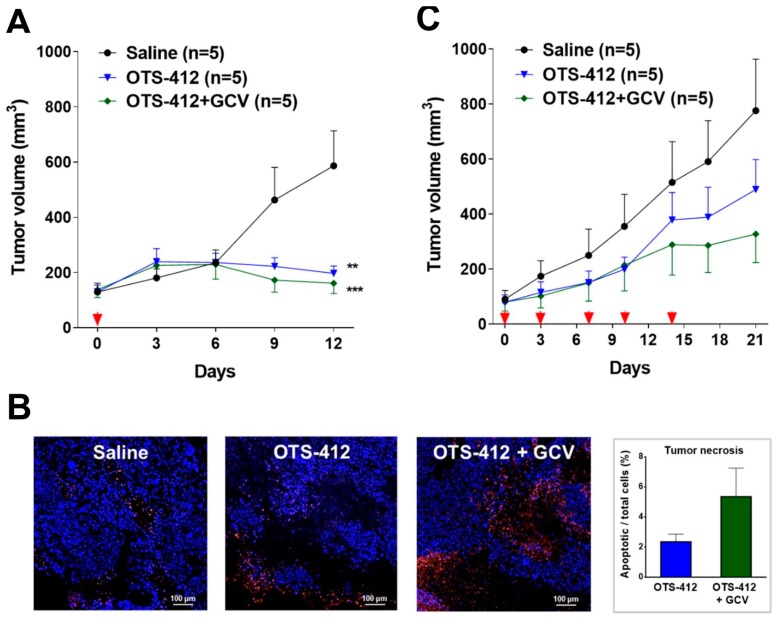
Anti-tumor efficacy of OTS-412, with or without ganciclovir (GCV), in two HCT 116 tumor-bearing mouse models. (**A**) (Study STG) Tumor volume measured from day 0 to day 12, post single-dose injection of saline, OTS-412 (day 0, intratumoral [IT], 1 × 10^6^ PFU), or OTS-412 + GCV (day 0, IT, 1 × 10^6^ PFU + once/day, day -3 to 11, intraperitoneal [IP], 25 mg/kg, respectively). ** *p* = 0.003, OTS-412 vs. Saline; *** *p* < 0.001, OTS-412 + GCV vs. Saline; *P* values were calculated using independent t-tests. (**B**) (Study STG) Terminal deoxynucleotidyl transferase dUTP nick end labeling (TUNEL) assay shows cancer cell apoptosis in one representative tumor mass from each treatment group in the abovementioned single-dose study; red signal represents apoptotic cancer cells and blue signal represents cell nuclei. The histogram shows the quantification of apoptosis in tumor tissue, which was not significantly different (*p* = 0.13). (**C**) (Study MVG) Tumor volume measured from day 0 to day 21, post multiple-dose saline injection, OTS-412 (on day 0, 3, 7, 10, and 14, IP, 1 × 10^8^ PFU), or OTS-412 + GCV (0, 3, 7, 10, and 14, IP, 1 × 10^8^ PFU + once/day, day 0 to 20, IP, 50 mg/kg, respectively). Scale bar: 100 µm.

## Data Availability

The data that support the findings of this study are available on request from the corresponding author. The data are not publicly available due to privacy or ethical restrictions.

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
