# Peer review of "Engineering and Characterization of Oncolytic Vaccinia Virus Expressing Truncated Herpes Simplex Virus Thymidine Kinase"

_cancers, 2020, doi:10.3390/cancers12010228_

Round 1
Reviewer 1 Report
The manuscript entitled: „Engineering and Characterization of Oncolytic Vaccinia Virus Expressing Truncated Herpes Simplex Virus Thymidine Kinase.” Suggests that the described new vaccinia virus variants exert higher oncolytic activity and specificity towards neoplastically transformed human cancer cells.
Although the experimental design appears adequate and the presented results convincing, there is a significant drawback, since there is a complete lack of information about the impact of these viruses on normal human cells. Indeed, as shown in Fig. 6 and suppl. Fig. 2, OTS-412 shows a significant reduction of replication in and killing capacitiy of mouse vs. human tumor cells, strongly suggesting that the changes introduced to the newly engineered viruses affect the host tropism rather than transformation specificity, in consequence also accounting for the tumor specificity in mouse models. To rule out this possibility, it is important to assess virus replication and toxicity in normal diploid human cell lines.
Also addressed in the Discussion, the rational to produce truncated HSVtk instead of TK- viruses is not entirely clear. Although, some properties of HSV-tk are shown in Fig. 1B (e.g. NBD and ABD), it is not apparent what residual function(s) are exerted by the truncated products and what effects the non-related C-termini could have. In addition, there appears no obvious rational to focus on OTS-412 and not on OTS-C1/OTS-C2. This should be clarified in Introduction and Results.
Author Response
Open Review
The manuscript entitled: „Engineering and Characterization of Oncolytic Vaccinia Virus Expressing Truncated Herpes Simplex Virus Thymidine Kinase.” Suggests that the described new vaccinia virus variants exert higher oncolytic activity and specificity towards neoplastically transformed human cancer cells.
Although the experimental design appears adequate and the presented results convincing, there is a significant drawback, since there is a complete lack of information about the impact of these viruses on normal human cells. Indeed, as shown in Fig. 6 and suppl. Fig. 2, OTS-412 shows a significant reduction of replication in and killing capacitiy of mouse vs. human tumor cells, strongly suggesting that the changes introduced to the newly engineered viruses affect the host tropism rather than transformation specificity, in consequence also accounting for the tumor specificity in mouse models. To rule out this possibility, it is important to assess virus replication and toxicity in normal diploid human cell lines.
Reply: Thank you very much for your comment about the normal cells, we have conducted an Incucyte assay in HUVEC-C cell line, to evaluate the cytotoxicity of OTS-412 (10 PFU/cell, which is 10-100 times higher than in human body) in normal human cell line. As you can see in the figure shown below, after 24 hours of infection, wild-type Wyeth VACV has caused considerably more cell apoptosis than control and OTS-412. As to the viral replication, the OTS-412 luciferase signal detected in HUVEC-C cells was 1/10 of that detected in HT-29 cancer cells. We have revised our method and result parts accordingly (Figures 1D and 1E).
Also addressed in the Discussion, the rational to produce truncated HSVtk instead of TK- viruses is not entirely clear. Although, some properties of HSV-tk are shown in Fig. 1B (e.g. NBD and ABD), it is not apparent what residual function(s) are exerted by the truncated products and what effects the non-related C-termini could have.
Reply: Thank you very much for your comment. The issue you point out is a limitation of this study as we mentioned in the end of the discussion, we have addressed this limitation as follow (line 394):
“2) Although the loss of pyrimidine catalyzing activity in OTS-412 was indirectly demonstrated through attenuated viral replication in various cancer cell lines, and through tumor selectively in animal models, we did not directly show the loss of function of the phosphorylation of thymidine (e.g., through an in vitro enzyme assay). 3) We did not clarify the molecular mechanism underlying “HSV-tk truncation losing thymidine sensitivity without affecting purine sensitivity”
However, it is important to clarify the function of the truncated part of HSV-tk in the future, even though it is not the primary objective of this current study.
In addition, there appears no obvious rational to focus on OTS-412 and not on OTS-C1/OTS-C2. This should be clarified in Introduction and Results.
Reply: Thank you very much for your comment. We added the data into the Supplemental Figure 1C, as follows:
Supplemental Figure 1C Viral replication and cytotoxicity of three viruses (OTS-412, OTS-C1, and OTS-C2) in NCI-H460 cancer cell line. Cells were seeded at 1.5×104 cells/well, virus concentration was 0.1 PFU/cell, incubation time was 48 hour. NT, negative control.
Reviewer 2 Report
The authors aim at engineering an oncolytic vaccinia virus (with a mutant HSV-tk transgene) with a feature of selective and enzymatic function towards GCV sensitivity and subsequently to assess anticancer efficacy and safety in combination.
Indeed, oncolytic viruses (OVs) are therapeutically useful anticancer viruses that will selectively infect and damage cancerous tissues without causing harm to normal tissues. OVs can access cells through binding to receptors on their surface or through fusion with the plasma membrane and establish a lytic cycle in tumors, while leaving normal tissue essentially unharmed.
The following points require further elaboration:
1) The authors stated in the introduction that ''Vaccinia virus (VACV) is one of the more preferred backbones for oncolytic virus engineering based on its long history of use as the routine vaccine for smallpox''. However, it would be worth to also state other oncolytic viruses like e.g. oncolytic adenoviruses and IMLYGIC (T-VEC/Talimogene Laherparepvec), a genetically engineered Herpes Simplex Virus, whi h is the first oncolytic virus approved for use in the United States and the European Union for the cancer patients (e.g. melanoma).
2) Would be also very beneficial for potential readers o know about the current oncolytic viruses limitations and made advances with a reference to the combinatory therapies (e.g. Garofalo M, JCR, papers).
3) How the dosage of VACV and GCV was selected. Did authors assess dose dependend effect of the GCV dose on virus replication?
4) The Supplemental Figure 4 presents survival in 4T1 tumor bearing mouse model. VACV did not show significant survival extension, suggesting that the therapy with the virus did not result in clinical efficacy. Was is related to los dosage selection?
5) Cytotoxic assay was carried out in order to assess the viral copy number of tested viruses and results are presented in the Supplemental Figure 2. However, what is missing is the group where virus was co-administered with the GCV.
6) I do not feel convinced that given statement ''The HSV-tk/GCV combination provides a safety mechanism for future clinical applications of OTS-412.'' was proven by the presented results. Therefore, the statement should be amended.
7) Discussion sections contains repetition from the Introduction.
8) The Figure 6C, the letter C overlap with the Incucyte live-cell imaging picture.
9) Did authors consider to perform the experiment utilizing humanized animal models, also looking into anti-cancer immune response development?
Author Response
Open Review
The authors aim at engineering an oncolytic vaccinia virus (with a mutant HSV-tk transgene) with a feature of selective and enzymatic function towards GCV sensitivity and subsequently to assess anticancer efficacy and safety in combination.
Indeed, oncolytic viruses (OVs) are therapeutically useful anticancer viruses that will selectively infect and damage cancerous tissues without causing harm to normal tissues. OVs can access cells through binding to receptors on their surface or through fusion with the plasma membrane and establish a lytic cycle in tumors, while leaving normal tissue essentially unharmed.
The following points require further elaboration:
1) The authors stated in the introduction that ''Vaccinia virus (VACV) is one of the more preferred backbones for oncolytic virus engineering based on its long history of use as the routine vaccine for smallpox''. However, it would be worth to also state other oncolytic viruses like e.g. oncolytic adenoviruses and IMLYGIC (T-VEC/Talimogene Laherparepvec), a genetically engineered Herpes Simplex Virus, whi h is the first oncolytic virus approved for use in the United States and the European Union for the cancer patients (e.g. melanoma).
Reply: Thank you very much, we agree with you that other oncolytic viruses should be mentioned as well. We have revised our introduction for T-VEC as follow (line 38):
“In 2015, the US Food and Drug Administration (FDA) and European Medicines Agency (EMA) approved their first oncolytic virus, T-VEC, a genetically engineered herpes simplex virus, for the treatment of advanced melanoma [2]. Meanwhile, many promising candidates engineered from different virus families have entered clinical trials [3-5]”
2) Would be also very beneficial for potential readers o know about the current oncolytic viruses limitations and made advances with a reference to the combinatory therapies (e.g. Garofalo M, JCR, papers).
Reply: Thank you very much, we have added the reference in the introduction as follow (line 42):
“In particular, the growing trend of combining oncolytic viruses with other cancer treatments, such as chemotherapies [6] and immune checkpoint inhibitors [7], has sparked significant interest in oncolytic virus research.”
3) How the dosage of VACV and GCV was selected. Did authors assess dose dependend effect of the GCV dose on virus replication?
Reply: Thank you for the important question. The VACV dosages were selected based on previous studies on other oncolytic viruses and GCV dosages used in other animal studies. We have inserted references into the method (line 542):
“The virus dosages used in this study (106 PFU for IT and 108 PFU for IP) were based on our previous studies on other OVVs [11,12]. The GCV dosages used in this study (25 and 50 mg/kg/day) were commonly adopted by many other animal studies [44,45], and according to a widely used human-animal drug dose translation equation [46], mouse dose of 50 mg/kg/day is equivalent to the recommended maintenance dosages for humans, which is around 5 mg/kg/day.”
As for the dose dependency of GCV, we evaluated GCV’s activity at 6 uM and 60 uM in our pilot study (the GCV concentration used in our in vitro study was 50 uM, equivalent to human concentration) and we observed no effect at 6 uM.
4) The Supplemental Figure 4 presents survival in 4T1 tumor bearing mouse model. VACV did not show significant survival extension, suggesting that the therapy with the virus did not result in clinical efficacy. Was is related to los dosage selection?
Reply: Thank you, it is a very good question. It is estimated that many factors can influence the efficacy of oncolytic vaccinia virus treatment, such as dosage, route of administration, tumor type, virus tropism, etc. It is such a complicated issue that may need to be clarified by many studies. In our unpublished data, additional modulation of host immune system can significantly improve the animal survival compared to oncolytic virus treatment alone.
5) Cytotoxic assay was carried out in order to assess the viral copy number of tested viruses and results are presented in the Supplemental Figure 2. However, what is missing is the group where virus was co-administered with the GCV.
Reply: Thank you for your comment. We have added a figure into the Supplemental Figure 1C as below:
Supplemental Figure 1C Viral replication and cytotoxicity of three viruses (OTS-412, OTS-C1, and OTS-C2) in NCI-H460 cancer cell line. Cells were seeded at 1.5×104 cells/well, virus concentration was 0.1 PFU/cell, incubation time was 48 hour. NT, negative control.
6) I do not feel convinced that given statement ''The HSV-tk/GCV combination provides a safety mechanism for future clinical applications of OTS-412.'' was proven by the presented results. Therefore, the statement should be amended.
Reply: Thank you for your comment, we agree that our original statement was not rigorous enough. We have revised this sentence as follow (line 599):
“The HSV-tk/GCV combination shows potential as a replication-control safety mechanism for future clinical applications of OTS-412.”
7) Discussion sections contains repetition from the Introduction.
Reply: We appreciate your comments, we have deleted the repetitive part in the discussion (line 286-299), and revised the introduction accordingly (line 76-84).
8) The Figure 6C, the letter C overlap with the Incucyte live-cell imaging picture.
Reply: Thank you for pointing it out for us, we have now made sure they did not overlap.
9) Did authors consider to perform the experiment utilizing humanized animal models, also looking into anti-cancer immune response development?
Reply: Thank you, this is a very important question, we have not yet conducted studies using humanized animal models, but we plan to conduct studies with ICI combination and immune modulation in future studies using the humanized mouse model.
Reviewer 3 Report
The authors describe the oncolytic vaccinia virus (OVV) OTS-412, in which the native thymidine kinase (tk) was replaced by a truncated version of the respective enzyme derived from herpes simplex virus (HSV). OTS-412 demonstrated similar replication potential and oncolytic capacity compared to tk-/- OVV. Its replication in tumor cell lines was reduced by half when ganciclovir was applied in vitro. In the RenCa and HCT116 tumor-bearing mouse models, the oncolytic virus was strictly confined to the tumor tissue after intratumoral and i.v. injection, and resulted in significantly reduced tumor growth, although the overall survival was not affected.
I congratulate the authors for addressing safety aspects of a virulent oncolytic virus via introducing a modification limiting its replication capacity. However, several questions need to be answered:
Major issues:
The authors used recombinant viruses with truncated HSV-1 tk. It is not clear to me whether this truncated protein is functional at all. The authors argue that it is not inhibited by aciclovir, while the effect of ganciclovir (at doses tenfold above the concentration used in humans) was very minor. Have the authors excluded toxic effects of high-dose ganciclovir? They should provide data of “GCV only” as control in Fig. 6 and 7B. Do the authors think that reducing viral replication by half is sufficient to minimize cytotoxic side effects of OVV. OTS-C1 and OTS-C2 reconstituted the 7G-stretch during replication. Did this modification affect the replication capacity and ganciclovir sensitivity of these viruses? Fig. 4: Please explain whether the mice died from recurrence of tumors within 20-30 days.Minor issues:
Introduction, l. 60: HSV-tk converts the prodrug GCV into GCV-monophosphate; the two additional phosphates are added via cellular kinases. Figure S1: please give more details in the figure legend. A: The authors have not included a size bar. Do the authors show two or four gels? Which is the correct size to be expected? Figure S1B: The marker lane is only poorly visible. Figure 1: please explain all abbreviations in the figure legend. Results, l. 95: please show effects of GCV on OTS-412, OTS-C1, and OTS-C2. The authors should discuss why the replication capacity of OTS-412 is tenfold lower in murine tumor cell lines.Author Response
Open Review
The authors describe the oncolytic vaccinia virus (OVV) OTS-412, in which the native thymidine kinase (tk) was replaced by a truncated version of the respective enzyme derived from herpes simplex virus (HSV). OTS-412 demonstrated similar replication potential and oncolytic capacity compared to tk-/- OVV. Its replication in tumor cell lines was reduced by half when ganciclovir was applied in vitro. In the RenCa and HCT116 tumor-bearing mouse models, the oncolytic virus was strictly confined to the tumor tissue after intratumoral and i.v. injection, and resulted in significantly reduced tumor growth, although the overall survival was not affected.
I congratulate the authors for addressing safety aspects of a virulent oncolytic virus via introducing a modification limiting its replication capacity. However, several questions need to be answered:
Major issues:
The authors used recombinant viruses with truncated HSV-1 tk. It is not clear to me whether this truncated protein is functional at all.
Reply: Thank you very much, one limitation of our study is that we did not study the molecular mechanisms underlying the loss of thymine kinase activity and preservation of purine sensitivity, as claimed in the study. We have added the limitations in the discussion as follows (line 394):
“2) Although the loss of thymidine catalyzing activity in OTS-412 was indirectly demonstrated through attenuated viral replication in various cancer cell lines, and through tumor selectively in animal models, we did not directly show the loss of thymidine phosphorylation activity (e.g., through an in vitro enzyme assay). 3) We did not clarify the molecular mechanism underlying HSV-tk truncation losing thymidine sensitivity without affecting purine sensitivity.”
The authors argue that it is not inhibited by aciclovir, while the effect of ganciclovir (at doses tenfold above the concentration used in humans) was very minor. Have the authors excluded toxic effects of high-dose ganciclovir? They should provide data of “GCV only” as control in Fig. 6 and 7B.
Reply: Thank you very much, we believe this is a very important question that we should have addressed more clearly. The in vitro studies with GCV treatment used 50 μM (~12.8 μg/mL) concentration, which was similar to human plasma Cmax which is around 10 μg/mL [1].
As for the in vivo dose used in the mouse studies (50 mg/kg), according to the previous publication [2], the mouse dose of 50 mg/kg is equivalent to human dose of ~5 mg/kg, we apologize for incorrectly addressing this (line 369), and have revised the method as follows (line 524):
“The GCV concentration used here, 50 μM, is similar to human plasma Cmax (maximum concentration) following an IV injection of GCV of 5 mg/kg, which is around 10 μg/mL [40].”
(line 542)
“The GCV dosages used in this study (25 and 50 mg/kg/day) were commonly adopted by many other animal studies [41,42], and according to a widely used human-animal drug dose translation equation [43], mouse dose of 50 mg/kg/day is equivalent to the recommended maintenance dosages for humans, which is around 5 mg/kg.”
In addition, although we did not directly test the cytotoxicity of GCV in vitro and in vivo, there has been sufficient evidence to support the insensitivity of human cells to high concentration of GCV (e.g., ~500uM) [3,4].
Do the authors think that reducing viral replication by half is sufficient to minimize cytotoxic side effects of OVV.
Reply: Thank you very much for your valuable comment. We have observed patients who died within weeks after oncolytic vaccinia virus injection in a previous clinical trial, and virus induced sepsis was suspected (we are going to publish the data soon). But it is difficult to reproduce such septic models because the tk gene deleted vaccinia virus (such as VVtk-) is highly attenuated. Even at 108 PFU, a dose hundreds of times higher than human dose, OTS-412 did not cause significant toxicity in mice. Therefore, it is difficult to evaluate GCV efficacy using mouse model, except using the virus copy number in tumor tissue to indicate the viral replication.
OTS-C1 and OTS-C2 reconstituted the 7G-stretch during replication. Did this modification affect the replication capacity and ganciclovir sensitivity of these viruses?
Reply: Thank you, it is a good question. In our study, OTS-412, -C1, and -C2, each have a modified HSV-tk gene, and showed similar replication with tk gene deleted vaccinia virus VVtk-(Figure 1C). In another in vitro study, OTS-412 showed significantly lower replication than wild-type Wyeth strain vaccinia virus (Supplemental Figure 2). However, we did not compare the replication of OTS-C1 and -C2 with the wild-type Wyeth strain vaccinia virus.
Fig. 4: Please explain whether the mice died from recurrence of tumors within 20-30 days.
Reply: Thank you for your comment, we only monitored overall survival, and did not record tumor volume nor did we conduct organ pathophysiological evaluation after the mice died, since our efficacy studies were conducted independently from the survival study. But the 4T1 mouse model has been well characterized, and is known as a very invasive/metastatic breast cancer model [5].
Minor issues:
Introduction, l. 60: HSV-tk converts the prodrug GCV into GCV-monophosphate; the two additional phosphates are added via cellular kinases.
Reply: Thank you for your advice, we have revised the introduction part as follow (line 70):
“Such combination takes effect relying on the rate-limiting step that HSV-tk mediates in converting the prodrug GCV into the cytotoxic GCV-triphosphate: HSV-tk coverts GCV into GCV monophosphate, the latter is subsequently converted to GCV-diphosphate and GCV-triphosphate by guanylate kinase and cellular kinase, respectively [14]. It is GCV-triphosphate that eventually halts DNA replication of both the virus and host cancer cell.”
Figure S1: please give more details in the figure legend. A: The authors have not included a size bar. Do the authors show two or four gels? Which is the correct size to be expected? Figure S1B: The marker lane is only poorly visible.
Reply: Thank you for your advice and questions. We have added more information into the figure legend for Supplemental Figure 1. In -1A, we showed four gels, but we took photos of two gels each time due to the large number of samples (120 plaques). This was the very first qualitative screening step; no markers were used. And in -1B, we used two plaques that were chosen from the 120 plaques for size evaluation, the clear marker was shown in Figure 1A.
Figure 1: please explain all abbreviations in the figure legend.
Reply: we appreciate your comments. We have added the definitions for the abbreviations in all figure legends (highlighted in each legend).
Results, l. 95: please show effects of GCV on OTS-412, OTS-C1, and OTS-C2.
Reply: Thank you for your good advice, we believe this is a very important question of our study. We have added a figure into the Supplemental Figure 1 as follow:
Supplemental Figure 1C Viral replication and cytotoxicity of three viruses (OTS-412, OTS-C1, and OTS-C2) in NCI-H460 cancer cell line. Cells were seeded at 1.5×104 cells/well, virus concentration was 0.1 PFU/cell, incubation time was 48 hour. NT, negative control.
The authors should discuss why the replication capacity of OTS-412 is tenfold lower in murine tumor cell lines.
Reply: Thank you for your insightful comment. It is known that different vaccinia virus strains replicate preferentially in different animal species. VV-tk delected Wyeth vaccinia viruses have been consistently found to replicate lower in murine cells than in human cells [6,7], the specific mechanisms are unclear, the species specific expression of INF-gamma receptors [8] may be one of the reasons.
Winston, D.J.; Baden, L.R.; Gabriel, D.A.; Emmanouilides, C.; Shaw, L.M.; Lange, W.R.; Ratanatharathorn, V. Pharmacokinetics of ganciclovir after oral valganciclovir versus intravenous ganciclovir in allogeneic stem cell transplant patients with graft-versus-host disease of the gastrointestinal tract. Biol Blood Marrow Transplant 2006, 12, 635-640, doi:10.1016/j.bbmt.2005.12.038. Reagan-Shaw, S.; Nihal, M.; Ahmad, N. Dose translation from animal to human studies revisited. FASEB J 2008, 22, 659-661, doi:10.1096/fj.07-9574LSF. Kobayashi, N.; Noguchi, H.; Westerman, K.A.; Matsumura, T.; Watanabe, T.; Totsugawa, T.; Fujiwara, T.; Leboulch, P.; Tanaka, N. Efficient Cre/loxP site-specific recombination in a HepG2 human liver cell line. Cell Transplant 2000, 9, 737-742, doi:10.1177/096368970000900525. Gentry, B.G.; Im, M.; Boucher, P.D.; Ruch, R.J.; Shewach, D.S. GCV phosphates are transferred between HeLa cells despite lack of bystander cytotoxicity. Gene Ther 2005, 12, 1033-1041, doi:10.1038/sj.gt.3302487. Pulaski, B.A.; Ostrand-Rosenberg, S. Mouse 4T1 breast tumor model. Curr Protoc Immunol 2001, Chapter 20, Unit 20 22, doi:10.1002/0471142735.im2002s39. Parato, K.A.; Breitbach, C.J.; Le Boeuf, F.; Wang, J.; Storbeck, C.; Ilkow, C.; Diallo, J.S.; Falls, T.; Burns, J.; Garcia, V., et al. The oncolytic poxvirus JX-594 selectively replicates in and destroys cancer cells driven by genetic pathways commonly activated in cancers. Mol Ther 2012, 20, 749-758, doi:10.1038/mt.2011.276. Cho, E.; Islam, S.; Jiang, F.; Park, J.E.; Lee, B.; Kim, N.D.; Hwang, T.H. Characterization of Oncolytic Vaccinia Virus Harboring the Human IFNB1 and CES2 Transgenes. Cancer research and treatment : official journal of Korean Cancer Association 2019, 10.4143/crt.2019.161, doi:10.4143/crt.2019.161. Alcami, A.; Smith, G.L. Vaccinia, cowpox, and camelpox viruses encode soluble gamma interferon receptors with novel broad species specificity. J Virol 1995, 69, 4633-4639.
Round 2
Reviewer 1 Report
I appreciate the author's effort to include effects of OTS-412 in HUVEC cells. However, I still believe that the evaluation of cytotoxicity in normal human cells is crucial to appreciate the improvement of novel Vector designs. In this regard a more quantitative Evaluation (e.g. MTT/LDH assays) including dose Response and/or time-course Experiments would have been preferable.
Reviewer 2 Report
The authors provided satisfactory replies and corrections.
Reviewer 3 Report
I have no further comments.